# Stand Out in Class: Investigating the Potential Impact of a Sit–Stand Desk Intervention on Children’s Sitting and Physical Activity during Class Time and after School

**DOI:** 10.3390/ijerph18094759

**Published:** 2021-04-29

**Authors:** Yu-Ling Chen, Keith Tolfrey, Natalie Pearson, Daniel D. Bingham, Charlotte Edwardson, Lorraine Cale, David Dunstan, Sally E. Barber, Stacy A. Clemes

**Affiliations:** 1School of Sport, Exercise and Health Sciences, Loughborough University, Loughborough LE11 3TU, UK; K.Tolfrey@lboro.ac.uk (K.T.); N.L.Pearson@lboro.ac.uk (N.P.); L.A.Cale@lboro.ac.uk (L.C.); S.A.Clemes@lboro.ac.uk (S.A.C.); 2National Institute for Health Research (NIHR) Leicester Biomedical Research Centre, University Hospitals of Leicester NHS Trust and University of Leicester, Leicester LE5 4PW, UK; ce95@leicester.ac.uk; 3Bradford Institute for Health Research, Bradford Teaching Hospitals Foundation Trust, Bradford BD9 6RJ, UK; Daniel.Bingham@bthft.nhs.uk (D.D.B.); Sally.Barber@bthft.nhs.uk (S.E.B.); 4Diabetes Research Centre, University of Leicester, Leicester LE5 4PW, UK; 5Baker Heart and Diabetes Institute, Melbourne, VIC 3004, Australia; David.Dunstan@baker.edu.au; 6Mary MacKillop Institute for Health Research, Australian Catholic University, Melbourne, VIC 3065, Australia

**Keywords:** standing desks, primary/elementary school, sedentary behaviour, children, after school, health inequalities, cluster randomised controlled trial

## Abstract

Sedentary behaviour (sitting) is a risk factor for adverse health outcomes. The classroom environment has traditionally been associated with prolonged periods of sitting in children. The aim of this study was to examine the potential impact of an environmental intervention, the addition of sit–stand desks in the classroom, on school children’s sitting and physical activity during class time and after school. The ‘Stand Out in Class’ pilot trial was a two-arm cluster randomised controlled trial conducted in eight primary schools with children from a mixed socioeconomic background. The 4.5 month environmental intervention modified the physical (six sit–stand desks replaced standard desks) and social (e.g., teachers’ support) environment. All children wore activPAL and ActiGraph accelerometers for 7 days at baseline and follow-up. In total 176 children (mean age = 9.3 years) took part in the trial. At baseline, control and intervention groups spent more than 65% of class time sitting, this changed to 71.7% and 59.1% at follow-up, respectively (group effect *p* < 0.001). The proportion of class time spent standing and stepping, along with the proportion of time in light activity increased in the intervention group and decreased in the control group. There was no evidence of any compensatory effects from the intervention after school. Incorporating sit–stand desks to change the classroom environment at primary school appears to be an acceptable strategy for reducing children’s sedentary behaviour and increasing light activity especially during class time. Trial registration: ISRCTN12915848 (registered: 09/11/16).

## 1. Introduction

The health benefits associated with engagement in regular physical activity across all age groups is unequivocal. However, advances in technologies and changes to our environments and lifestyles have typically reduced the requirement to incorporate regular activity, or movement-related behaviours, into our daily routines. In children, regular physical activity is associated with improved learning and attainment, cardiovascular fitness and mental health, and also a healthy weight status [1]. Despite these known benefits, physical inactivity in children in the UK remains a major public health concern. Nationally representative data from England suggest that only 21% of children aged 5–15 years engage in sufficient levels of physical activity to benefit their health [2]. Coupled with low levels of physical activity, sedentary behaviour is ubiquitous within all settings of daily life. Sedentary behaviour is defined as ‘any waking behaviour characterised by an energy expenditure ≤1.5 metabolic equivalents (METs) while in a sitting, reclining or lying posture [3]. Sitting is the most prevalent behaviour exhibited during waking hours, reportedly accounting for over 65% (~7.5 h/day) of waking time in a sample of children from the UK [4].

Whilst evidence of the associations between sedentary time and increased risk of adiposity/weight gain and clustered cardiometabolic risk in children is largely restricted to the study of screen time [5], engagement in sedentary behaviours has been shown to increase across key transitions in children’s lives (e.g., from primary to secondary school) [6], and track into adolescence [7] and adulthood [8]. Reducing children’s sitting time may, therefore, be important for the primary prevention of chronic diseases in adulthood, given the links observed between sedentary behaviour and numerous adverse health outcomes in adults [9,10]. In parallel to reducing total sitting time, breaking up and reducing prolonged periods of sitting have been highlighted as potential strategies to limit the detrimental effects of sedentary behaviour, and have been shown to lead to positive effects on metabolic outcomes and cognitive function in children and young people [11,12]. The development of an increased cardiometabolic health risk profile in some ethnic groups is also evident during childhood [13]. For example, higher levels of adiposity, glycated haemoglobin, fasting insulin and triglycerides, with reduced high-density lipoprotein-cholesterol, have been observed in British South Asian heritage children compared with White British children [14,15]. Higher volumes of sedentary time and lower levels of physical activity have also been observed in South Asian heritage primary/elementary school-aged children (aged 6–11 years) compared with White British children [16,17,18]. Given the links between sedentary behaviour and cardiometabolic risk [5], early interventions in such at-risk groups may help to reduce health inequalities later in life [19].

As children spend approximately half of their waking hours at school, this environment has been highlighted as an important setting to target the promotion of health enhancing lifestyle-related behaviours [20,21,22], particularly in relatively deprived locations [19]. Children growing up within lower socio-economic environments experience a range of adverse health and developmental outcomes [23,24] and exhibit a greater risk of developing a variety of chronic diseases in adulthood compared to those from less deprived areas [25]. Furthermore, ethnic minority children such as those of South Asian heritage (e.g., Pakistani, Indian, Bangladeshi heritage) tend to be overrepresented in the most deprived neighbourhoods in England [26]. School classroom-based interventions, therefore, have the potential to address health inequalities when they are accessible to all attending children [27]. Yet, the school classroom environment typically promotes high volumes of sitting. For example, in samples of children (aged 9–10 years) from primary schools in northern England, we have observed that at least 70% of class time is spent sitting [28].

To target this traditionally sedentary environment, there has been increased interest in the use of sit–stand desks (desks which enable children to alternate their posture between sitting and standing) within the classroom setting. Sit–stand desks, when incorporated into the classroom environment, have shown potential for breaking up and reducing children’s sitting time and increasing movement. To date though, most studies within this area have been limited to quasi-experimental or crossover trials with relatively short intervention periods (e.g., <12 weeks) or small-scale single-school pilot studies [29,30,31,32]. Recent studies from a range of countries have supported these earlier findings and demonstrated that sit–stand desks within the classroom leads to beneficial changes (reductions) in children’s class/school-time spent sitting [33,34,35,36,37] or total daily time spent sedentary [19,38,39]. However, this evidence remains largely limited to studies located within single school settings [34,35,36,37,38], and little evidence is currently available from randomised controlled trials (RCTs) [19,39]. Further, whilst studies appear to demonstrate that school-based sit–stand desk interventions do not lead to negative changes in behaviour outside of school, such as compensatory increases in sitting during leisure time [33,35,36,37], limited evidence has been collected from relatively higher-risk populations living within more deprived settings [33], where higher levels of sedentary behaviour and physical inactivity are more prevalent [16,17,18].

This paper utilises device-based sitting and physical activity data collected from the ‘Stand Out in Class’ pilot cluster RCT [19,27,40], which examined the feasibility and preliminary effectiveness of a classroom-based sit–stand desk intervention. This intervention incorporated an environmental variable, the sit–stand desks, which modified both the physical and social environments of the classroom. The trial took place within primary schools located in the City of Bradford, chosen as the study setting given its ethnic composition (predominantly South Asian heritage and White British) and high levels of deprivation, health inequalities and childhood morbidity [41]. Whilst the primary feasibility-related outcomes and the preliminary effectiveness of the intervention on total school-day sitting have been reported elsewhere [19,40], further interrogation of the data are required to explore the potential impact of the intervention on children’s sedentary behaviour and physical activity across different contexts/environments. Increasing our understanding of how this classroom-based intervention may impact sitting (including duration and bouts) and activity behaviours within and outside of this environment will be essential to inform the planning of future fully powered effectiveness trials. Therefore, the aim of the present article is to provide a comprehensive understanding of the potential impact of a classroom sit–stand desk intervention on children’s sitting and physical activity behaviours within the classroom and after school settings.

## 2. Materials and Methods

### 2.1. Study Design

The protocol for this pilot trial has been reported elsewhere [27,40]. Briefly, the Stand Out in Class trial was a school-based two-arm pilot cluster RCT. The intervention was delivered within the classroom environment. Baseline measurements (November 2016) preceded randomisation (December 2016), and an identical set of measures were taken from all participants in July (2017). Schools were stratified according to predominant pupil ethnicity and randomly assigned to one of two conditions: (1) six manually adjustable sit–stand desks incorporated into the classroom environment (intervention condition), or (2) current practice (control condition) [19]. The reporting of this trial follows the CONSORT extension statement for cluster trials [42]. Ethical approval for this study was obtained from Loughborough University’s Ethical Advisory Committee (reference: R16-P027).

### 2.2. School and Pupil Eligibility

A selection of targeted government funded primary schools located in the City of Bradford with at least 25 pupils in a year 5 class were invited to take part. Schools with predominantly South Asian heritage pupils (>50%) and predominantly White British pupils (>50%) were specifically targeted for inclusion in the study [27,40]. Information on the ethnic composition of the schools’ pupil population was determined using local school census data [27]. Eight eligible schools, four with predominantly South Asian heritage pupils and four with predominately White British pupils were recruited. Consenting schools were asked to nominate a year 5 class and were provided with invitation packs for the parents/guardians of children within these classes. All children within the participating classes were eligible to take part in the intervention. Parental written informed consent and child assent were required for participation.

### 2.3. The Intervention

The Stand Out in Class intervention included the provision of six height-adjustable sit–stand desks (LearnFit, Ergotron Inc., USA) placed in the nominated year 5 classroom (replacing three standard desks sitting 6 children) of each of the four intervention schools for two school terms, spanning 4.5 months (February–July 2017) [27,40]. The research team provided training to teachers and pupils on sit–stand desk use. The training also included a presentation on the benefits of regular physical activity and reductions in sedentary time. In addition, teachers received a Professional Development Manual (available on request) and a series of nudging prompt cards containing information on the health benefits of reducing prolonged sitting and on correct posture when standing at the desks. The training, manual and prompt cards focused on encouraging correct adoption of the intervention targeting key barriers and facilitators to sit–stand desk use, informed by our previous work [31,33,40] from the Capability, Opportunity, and Motivation-Behaviour model (COM-B) within the Behaviour Change Wheel [43] and the Theoretical Domains Framework [44] (e.g., self-efficacy, motivation and knowledge). Standardised behaviour change techniques (e.g., goal setting, instruction) [45] were also applied during the training with teachers and pupils [19,27,40]. The research team supported teachers in modifying the classroom environment and developing a classroom rotation plan to ensure all children in their class were exposed to the sit–stand desks for at least one hour per day on average across the week. A group contract to encourage each other to stand more was also signed by the children at the beginning of the intervention. Stools or chairs remained in the classroom and children were free to choose whether they sat or stood when using the sit–stand desks [19,27,40].

### 2.4. The Usual Practice Control Arm

The four schools assigned to the control condition were asked to continue with their usual practice and took part in the study measurements at the same two time points using the same measures as those in the intervention condition. Upon completion of the study, control schools were offered a report summarising the collected data of their pupils [27,40].

### 2.5. Measures

Demographic data such as sex, age and ethnicity were provided to the research team from the schools own census data and captured using self-report questionnaires. Anthropometric data including children’s stature and body mass were measured by trained research staff and BMI was calculated (kg/m^2^). Sitting/sedentary behaviour and physical activity were captured for 7 consecutive days (24 h/day) using an activPAL micro accelerometer (PAL Technologies, UK) worn on the anterior aspect of the right thigh, and an ActiGraph GT3X+ accelerometer (ActiGraph, Pensacola, FL, USA) worn on the waist, at both baseline and follow-up. The activPAL was chosen as an established measure of posture, whilst the ActiGraph was chosen as an established measure of physical activity. Participants were provided with a brief diary during each monitoring period in which they were asked to document time in bed and any periods of non-wear for either device.

The activPAL, shown to provide a valid measure of posture in children [46], was waterproofed by wrapping it within a nitrile sleeve and hypoallergenic Hypafix (BSN Medical) dressing. It was attached to the thigh using a second piece of Hypafix (BSN Medical) dressing. All activPALs were initialised and downloaded using manufacturer proprietary software (activPAL Professional v.7.2.32, PAL Technologies Ltd., Scotland, UK.). The activPAL data were cleaned, processed and summarised using the freely available ProcessingPAL application (https://github.com/UOL-COLS/ProcessingPAL, (accessed on 12 April 2021) version 1.1, University of Leicester, (Leicester, UK)). Periods of prolonged non-wear, and time in bed and invalid data were excluded from the analyses using an automated algorithm within the ProcessingPAL application [47], and supplemented with cross-checking against participants’ diary entries by the authors (YLC and NP). During data processing, the ProcessingPAL application attributed each continuous period of sitting/lying during waking hours as a sitting bout according to its automated algorithm [47].

ActiGraphs were initialised to record data at 60 Hz. The devices were initialised and downloaded using ActiLife version 6.13.3, and the data (reintegrated into 15 s epochs) were processed using specifically developed and commercially available software (KineSoft version 3.3.20, Loughborough, UK) [19]. This commercially available software allowed data to be processed according to school bell times and is widely used to process ActiGraph data collected from children [48]. Time spent sedentary, in light physical activity (light PA) and in moderate-to-vigorous physical activity (MVPA) throughout the day were extracted from the ActiGraph data using the Evenson cut-points [49]. A blanket removal of sleep time between 23:00 and 05:59 was undertaken when processing these data. However, to identify periods of sleep and/or non-wear occurring outside of this time period (i.e., after 06:00 and before 23:00), the 3-axis acceleration data from the ActiGraph were used to detect periods of no movement. If these periods exceeded 20 min of zero counts, then this additional period was excluded as non-wear/sleep time [27,40]. Due to the exploratory nature of this study, children were included in the analyses if they had worn the activPAL and/or ActiGraph for at least 8 hours on at least 1 weekday at baseline and follow-up [27,40].

In order to identify class time and after school time on school days, the timetable and bell times of each school during the studied period were obtained from all eight schools. Class time data were extracted as the time periods between the first and last bell, with the removal of data collected during the morning break and lunch time. After school data were identified as the time following the last bell time to 23:00. These principles were applied when processing both the activPAL and ActiGraph data (ActiGraph, Pensacola, FL, USA).

### 2.6. Statistical Analyses

All statistical analyses were conducted using SPSS v.26 (SPSS Inc., Chicago, IL, USA). Comparisons of the characteristics of the participants at baseline between the intervention and control groups were conducted using independent t-tests for continuous variables (i.e., age and BMI), and chi-square tests for categorical variables (i.e., sex and ethnicity). Given the pilot nature of the present study and the consideration that the trial was not powered to determine effectiveness, the analyses conducted should be treated as preliminary. The analyses described below were undertaken to explore the potential of the intervention to impact upon sedentary and active behaviours within different contexts. The preliminary analysis applied individual children as the unit of analysis with sitting, standing, stepping, number of steps (derived from the activPAL) and sedentary time, light PA, and MVPA (derived from the ActiGraph) during different windows of time (in class and after school time) being the variables of interest. The percentages of times spent in each behaviour relative to the total wear time were calculated and included as variables. ANCOVA models weighted by school were performed to examine the potential effects of the intervention on the measured behaviour. The baseline value of the outcome variable of interest was included as the covariate and the group (intervention versus control) effects were reported. Cohen’s d was calculated for the absolute paired effect sizes. Effect sizes smaller than 0.20 were considered as trivial; 0.20 to <0.50 small; 0.50 to <0.80 medium; and effect sizes ≥0.80 were large [50].

## 3. Results

A total of 176 children from eight clusters (schools) completed both baseline and follow-up measurements, representing 97% of those with parental consent at baseline. Detailed results of the descriptive analyses and all trial feasibility related outcomes are reported elsewhere [19,40]. The descriptive characteristics of the sub-samples (those with valid activPAL (*n* = 108) and ActiGraph (*n* = 145) data at baseline and follow-up) included in the analyses of the current paper are presented in Table 1. There were no significant differences in age (*p* > 0.05), sex distribution (*p* > 0.05) and BMI (*p* > 0.05) between participants in the intervention and control group at baseline. However, chi-square tests revealed that there were significantly more South Asian Heritage children in the control arm, and more white British children in the intervention arm (*p* < 0.001). The overall sample mean BMI was 18.3 kg/m^2^ which is categorised as normal weight (between the 2nd and 91st centiles) for children between 9 and 10 years old [51]. Overall, 29.5% of participating children had a BMI above the 91st centile. 

### 3.1. Device-Derived Time Spent Sitting, Standing, Stepping, in Light PA and MVPA

The participating children spent an average of 378 min/day in school, which included an average of 301 min/day in the classroom. Their average device wear time in the after school period was 462 min/day. The weighted means and standard deviations (SD) of the device-measured behaviour at baseline and follow-up are reported by group in Table 2. From baseline to follow-up, the mean activPAL wear time during both class and after school time decreased in the control group, but increased in the intervention group. The mean ActiGraph wear time during class time increased slightly in the control group, but decreased in the intervention group; conversely, mean after school ActiGraph wear time decreased similarly in both groups. Due to the between group differences in device wear time, the absolute mean values of time spent in each behaviour were reported to provide the descriptive behavioural information of the participants only, with the preliminary inferential analyses utilising the data relating to the proportion of time spent in each behaviour in the following subsections.

### 3.2. The Percentage of Time Spent in Device-Measured Behaviour Relative to Total Wear Time

#### 3.2.1. Class Time

Between group comparisons of the proportions of wear time spent sitting, standing and stepping, and in light PA and MVPA did not differ significantly between groups at baseline, with the exception of the proportions of wear time (%) spent stepping (*p* = 0.001, d = 0.15) and in MVPA (*p* < 0.001, d = 0.02), with the control group accumulating more time in these behaviours during class time than the intervention group. The between group comparisons of follow-up proportions of wear time (%) spent sitting, standing, stepping (derived from the activPAL) and in sedentary behaviour, light PA, and MVPA (derived from the ActiGraph), after controlling for baseline values, were all significant (*p* < 0.001) during class time (Table 3). The difference in the proportion of time spent sitting during class time between the control (71.7%) and intervention (59.1%) groups at follow up was large (d = 1.10). Similarly, the follow up significant differences in the proportion of time spent standing and stepping between groups during class time (Table 3) had large effect sizes, in favour of the intervention arm (d = 0.84). The significant differences in behaviour measured by the ActiGraph were smaller than those described above for the activPAL during class time (see F ratios in Table 3); however, they were still indicative of an effective intervention that resulted in a moderate reduction in sedentary time (d = 0.51) and small (d = 0.47 & 0.33) increases in light PA and MVPA, respectively, in the intervention group compared to the control group.

#### 3.2.2. After School

Between group comparisons of the proportions of wear time spent sitting, standing and stepping, and in light PA and MVPA after school did not differ significantly (*p* > 0.05) between groups at baseline. The between group comparisons of follow-up proportions of wear time (%) spent sitting, standing, stepping (derived from the activPAL), and in sedentary behaviour, light PA, and MVPA (derived from the ActiGraph), after controlling for baseline values, were nearly all significant (*p* < 0.001) during the after school period (Table 3). Only the proportion of wear time spent in MVPA was non-significant between groups (*p* = 0.621) (Table 3). The differences in follow-up activPAL-derived sitting, standing and stepping proportions between groups were small or trivial (d = 0.24, 0.12 and 0.28, respectively). Similarly, the between group follow-up differences for the proportion of time spent in ActiGraph-derived sedentary behaviour (d = 0.36), light PA (d = 0.47) and MVPA (d = 0.00) were also small or trivial.

### 3.3. Sitting Bouts

The potential effect on the total number of sitting bouts during class time between groups was similar (*p* = 0.245) with a small between group effect (d = 0.23) (Table 4). The number of short sitting bouts at follow-up was significantly different (*p* = 0.022) between the control (38.2) and intervention (42.5) groups, although the effect was small (d = 0.31). The number of medium-to-long sitting bouts was also significantly different (*p* < 0.0005), with a large effect (d = 0.82) in favour of the intervention arm (Table 4). However, the real difference of 0.7 bouts is, again, unlikely to be meaningful. When the follow-up short and medium-to-long sitting bouts were expressed as proportions of the total sitting bouts, the between group moderate (d = 0.61) differences were significant (*p* < 0.001). The intervention group had a greater proportion in short compared with medium-to-long sitting bouts (89.2 vs. 10.8%) than the control (79.0 vs. 21.0%) group at follow-up, although the number of sit-to-stand transitions during class time were similar between groups (*p* = 0.135; d = 0.26). Examination of the after school bouts of sitting time revealed similar statistical trends to class time, albeit with less striking probabilities and notably smaller effects with them all being trivial (d = 0.11 to 0.17).

## 4. Discussion

The purpose of this study was to provide a comprehensive understanding of the potential impact of a sit–stand desk intervention on children’s sitting and physical activity within both the classroom and afterschool settings. The findings suggest that the incorporation of six sit–stand desks in the classroom physical environment and social environment (e.g., teachers’ support, education session and social contract, etc.) have the potential to reduce children’s sitting time, and increase standing and stepping time, during class time in the intervention group, relative to the control group. Furthermore, reductions in time spent in medium to long sitting bouts (≥30 min) during class time were more likely to be seen in the intervention arm, relative to the control group, although the absolute difference in the number of medium-to-long sitting bouts between groups was small, questioning the meaningfulness of this finding. Differences in sitting, standing and stepping behaviours, and light physical activity were also observed between groups after school but these differences were small or trivial, suggesting the sit–stand desk intervention did not measurably affect the behaviour outside the classroom. In general, findings of this pilot cluster RCT are in line with the findings of the previous pilot study conducted by our research team, which pioneered the “six sit–stand desks” approach [52]. Whilst a greater reduction in sitting was observed in the previous pilot study, this earlier evidence was generated from a controlled trial completed in a single primary school [52]. The current pilot cluster RCT provided further evidence from a more robust study design with data collected from children across eight schools. Similar reductions in activPAL-measured sitting time (–26 mins/day) during school hours were reported in an RCT of a sit–stand desk intervention in Belgian primary school children; however, this study incorporated just three sit–stand desks in classrooms as opposed to six used herein [39].

Similar findings have also been reported in studies that have applied a whole classroom approach, allocating one sit–stand desk to each pupil. Pilot controlled trials in both the UK and Australia have reported significant reductions in sitting time when every pupil is provided with a sit–stand desk [33,52]. In these studies, reductions in classroom sitting time have ranged from –44 to –53 min/day (during class time). A recently published crossover trial in Australia providing one sit–stand desk for each child during the intervention condition (lasting 21 school days), observed reductions in sitting (–28 mins/day during school time) in this condition relative to the control condition [38]. Another crossover trial conducted in an American elementary school found that whilst using standing desks as the environmental intervention, children increased their sedentary time in this condition (by 2.4%), although this increase was smaller than the increase in sedentary time observed in the traditional desk condition (6.5%) over nine weeks [34]. The same study also found that the effect of the stand-biased desk intervention on sedentary behaviour was more effective among those students who were more sedentary at the beginning of the school year [34]. Interventions using a whole classroom approach are likely to reduce demands on teachers, in terms of negating the requirement to rotate children around the class to ensure equal exposure to the sit–stand desks, as used herein. However, relative to the intervention approach used in the present study (6 sit–stand desks), potential barriers highlighted against adopting a whole classroom approach include increased costs and potential space restrictions (depending on the size of the sit–stand desks versus traditional desks) [42,43].

Further analyses on sitting bouts revealed that children in the intervention group tended to reduce their number of medium to long sitting bouts during class time. Yet, the change in the number of medium-long sitting bouts was small and possibly not behaviourally meaningful. Furthermore, no significant changes in the number of breaks in sitting (i.e., sit-to-stand transition) were found. These findings are different from the large reductions in the number of sit-to-stand transitions seen in previous studies incorporating standing workstations (i.e., non-adjustable standing desks) in primary/elementary school classrooms in New Zealand [53,54]. One possible explanation of the non-significant finding in our study could be that the participants generally did not accumulate time in medium to long sitting bouts at baseline (i.e., total number of medium to long sitting bouts for the control and intervention groups were: 1.2 and 0.8, respectively), suggesting there is a floor effect and breaking up periods of prolonged sitting (>30 min) within the classroom setting may not be an issue for children of this age. Moreover, there was no specific “target time” prescribed for standing to the participants in our intervention with children in the intervention group simply being encouraged to stand more autonomously. One Australian study designed their intervention specifically to create the social norm of regularly breaking up sitting every 15 min and found a significant reduction in the number of prolong sitting bouts [55]. Although the study was conducted in a secondary school, the differences in approach could likely explain the differences between their findings and those of the current study. Nevertheless, the existing evidence of the health benefits of breaking up sitting time are based on different “break frequencies”. Whilst the beneficial effects of regularly breaking up sitting in adults has been widely reported [56], limited research has investigated the health effects of breaking up sitting in children, and that which has been conducted has adopted different ‘break’ protocols, with breaks occurring either every 20 or 30 min [11,12].

The previously published feasibility-related outcomes paper from this pilot cluster RCT reported that the intervention showed potential to reduce children’s total daily sitting time, in the intervention arm, by 30 min per day on weekdays compared to the control arm [19]. Whilst this pilot trial was not powered to determine effectiveness, the preliminary findings of the present paper extend those of our earlier paper and suggest that the changes in sitting time seen in the intervention group, relative to the control group, are the result of changes in sitting occurring within the classroom context. Limited meaningful differences in sedentary and active behaviours were observed between groups after school. This reinforces the notion that compensatory changes in sitting and physical activity did not appear to transcend from the classroom to the after school period [35,36,37]. While the present findings may suggest that no compensatory effects on children’s after school behaviour occurred, children in both the intervention and control groups accumulated more time in prolonged sitting bouts after school at baseline and follow-up. This finding could be related to the high levels of deprivation in the area where this pilot cluster RCT took place, where children’s environments may not be conducive to physical activity outside due to safety concerns or lack of facilities. These observations suggest that future interventions should target sedentary behaviour in both the classroom and after school environments (e.g., home environment).

There was a significant difference in the proportion of ethnic groups within the intervention and control arms. As the schools were randomised into the intervention and control arms, the results could be based on chance. However, further research with a larger sample size is required to understand the roles ethnicity and culture play in school-based sitting and physical activity interventions. A key strength of this study was the inclusion of a mixed social gradient of the school setting. Half of the participating schools were located in an area with high levels of deprivation and the majority of the participating children were of South Asian heritage, a minority group more likely to live within the most deprived 10% of neighbourhoods in the UK [26]. The findings of this study showed that the classroom-based intervention could reach all school children from different socioeconomic backgrounds and ethnicities and provide a potential valuable environment within which to target reductions in the sitting time of those most disadvantaged. This kind of proportionate universal approach is also suggested to be more likely to reduce health inequalities [57], and is aligned with the recent policies and guidelines published by the World Health Organisation [58,59].

Another strength of this pilot cluster RCT was the use of two devices to measure children’s behaviour and movement. Greater potential effects were seen from the activPAL data, likely because this device provides a direct measure of posture and, thus, accurately distinguishing sitting from standing; in contrast, the absence of movement is used as a proxy to estimate sedentary time with waist-worn ActiGraph data. The sit–stand desk intervention probably replaced sitting with standing, and periods of standing still were misclassified as sedentary time by the ActiGraph. This might also explain why some previous sit–stand desk studies in children have found smaller changes in sedentary time, or no change in light physical activity, whilst using the ActiGraph as their primary outcome measure [34,37]. The activPAL, or similar thigh-worn devices, which directly measure posture, are recommended as the primary outcome measure in future work of this nature. However, the compliance rate was 63% for the activPAL, which is much lower than the 83% compliance rate seen for the ActiGraph [19,40]. Further research should, therefore, examine different attachment options for the activPAL in children to improve compliance.

## 5. Conclusions

The findings suggest that the incorporation of six sit–stand desks within the classroom environment demonstrate the potential to reduce children’s sitting time in this setting. Limited meaningful changes in activity and sitting behaviours were seen outside of school, adding to the evidence that compensatory increases in sitting are unlikely. This relatively cost-efficient and more practical intervention approach, incorporating six sit–stand desks as opposed to one per pupil into classrooms, is suggested as a feasible intervention approach going forwards. The findings from this study warrant validation in a fully powered effectiveness trial with a longer intervention and follow-up period, the findings of which could lead to policy changes surrounding the provision of sit–stand desks within the classroom environment.

## Figures and Tables

**Table 1 ijerph-18-04759-t001:** Descriptive characteristics of the overall sample and the sub-samples who provided valid activPAL and ActiGraph data at both baseline and follow-up assessment sessions.

	Overall	Total Sample		Sub-Sample with Valid activPAL Data		Sub-Sample with Valid ActiGraph Data	
		CON	INT	CON	INT	CON	INT
Clusters (*N*)	8	4	4	4	4	4	4
Participants (*N*)	176	90	86	53	55	74	71
Age * (years)	9.3 ± 0.5	9.3 ± 0.5	9.3 ± 0.4	9.8 ± 0.4	9.8 ± 0.3	9.8 ± 0.3	9.8 ± 0.3
Sex							
Boys, *N* (%)	98 (56)	50 (56)	49 (56)	34 (64)	28 (51)	39 (53)	44 (62)
Girls, *N* (%)	78 (44)	40 (44)	37 (44)	19 (36)	27 (49)	35 (47)	27 (38)
Ethnicity, *N* (%)							
SAH	85 (48)	59 (66)	26 (30)	36 (68)	18 (33)	48 (65)	24 (34)
WB	63 (36)	18 (20)	45 (52)	13 (25)	35 (64)	22 (30)	42 (59)
Other, *N* (%)	28 (16)	13 (14)	15 (18)	4 (7)	2 (3)	4 (5)	5 (7)
BMI * (kg/m^2^)	18.3 ± 3.6	18.2 ± 4	18.2 ± 3.3	17.8 ± 3.5	18.1 ± 3.4	18.3 ± 3.8	18.0 ± 3.1

CON: Control; INT: Intervention; * Mean ± SD SAH: South Asian heritage; WB: White British; BMI: Body Mass Index (weight [kg]/height [m^2^]).

**Table 2 ijerph-18-04759-t002:** Device-measured behaviour (mean ± SD) at baseline and follow-up.

	Control Group		Intervention Group	
**During Class Time**	**Baseline**	**Follow-Up**	**Baseline**	**Follow-Up**
***activPAL data (n = 108)***				
Wear time (min)	311.8 ± 17.1	296.5 ± 36.2	273.4 ± 26.8	285.6 ± 15.5
Sitting time (min)	204.3 ± 47.9	212.2 ± 38.4	182.6 ± 26.7	169.2 ± 39.2
Standing time(min)	68.2 ± 30.6	54.5 ± 20.8	62.8 ± 23.2	75.3 ± 27.3
Stepping time(min)	39.3 ± 15.7	29.8 ± 15.0	28.0 ± 9.2	41.1 ± 15.7
Class step counts (N)	3016 ± 1280	2222 ± 1364	2094 ± 659	3178 ± 1287
***ActiGraph data (n = 145)***				
Wear time (min)	308.0 ± 9.1	312.7 ± 8.6	285.7 ± 14.1	277.5 ± 28.1
SB time (min)	179.7 ± 33.8	188.3 ± 33.9	166.9 ± 33.3	148.8 ± 31.6
Light PA time (min)	119.0 ± 29.0	118.3 ± 31.6	114.1 ± 28.4	121.7 ± 35.3
MVPA time (min)	9.3 ± 6.0	6.1 ± 7.8	4.8 ± 3.6	7.0 ± 4.8
**After School Time**	**Baseline**	**Follow-Up**	**Baseline**	**Follow-Up**
***activPAL data (n = 108)***				
Wear time (min)	391.1 ± 52.1	377.4 ± 71.3	340.6 ± 51.8	374.2 ± 54.8
Sitting time (min)	268.6 ± 47.5	216.6 ± 71.8	221.9 ± 39.7	224.1 ± 49.0
Standing time(min)	68.1 ± 24.0	78.7 ± 30.3	71.4 ± 22.6	76.1 ± 25.3
Stepping time(min)	54.4 ± 25.6	82.1 ± 36.5	47.3 ± 15.8	74.0 ± 31.7
Step counts (N)	4096 ± 2134	6290 ± 2871	3646 ± 1314	5856 ± 2618
***ActiGraph data (n = 145)***				
Wear time (min)	385.2 ± 70.6	359.2 ± 109.8	383.3 ± 52.1	359.5 ± 88.1
SB time (min)	195.9 ± 49.5	153.5 ± 66.7	198.8 ± 40.8	172.0 ± 52.1
Light PA time (min)	175.4 ±44	181.2 ± 69.4	169.7 ± 38.7	164.9 ± 49.7
MVPA time (min)	14.0 ± 9.3	24.4 ± 24.5	14.8 ± 8.7	22.6 ± 16.8

SD: Standard Deviation; SB: Sedentary Behaviour; PA: Physical Activity; MVPA: Moderate to Vigorous Physical Activity.

**Table 3 ijerph-18-04759-t003:** Percentage of wear time spent in device-measured behaviour (mean ± SD) and the preliminary effect of the treatment (intervention).

	Control		Intervention		ANCOVA * Group Effect	
**During Class Time**	**Baseline**	**Follow-Up**	**Baseline**	**Follow-Up**	**F**	***P***
***activPAL data***						
% time spent sitting	65.3 ± 13.6	71.7 ± 9.6	67.1 ± 9.7	59.1 ± 12.9	187.4	<0.001
% time spent standing	22.0 ± 10.0	18.3 ± 6.7	22.7 ± 7.6	26.5 ± 9.7	124.7	<0.001
% time spent stepping	12.7 ± 5.2	10.0 ± 4.8	10.1 ± 3.0	14.4 ± 5.7	114.1	<0.001
***ActiGraph data***						
% time spent in SB	58.2 ± 10.3	60.3 ± 11.1	58.3 ± 10.7	54.1 ± 13.2	62.4	<0.001
% time spent in LPA	38.7 ± 9.7	37.8 ± 9.8	40.1 ± 10.2	43.3 ± 13.2	38.6	<0.001
% time in MVPA	3.0 ± 2.0	1.9 ± 2.5	1.6 ± 1.2	2.6 ± 1.7	41.5	<0.001
**After School Time**	**Baseline**	**Follow-Up**	**Baseline**	**Follow-Up**	**F**	***P***
***activPAL data***						
% time spent sitting	68.8 ± 9.0	57.0 ± 14.9	65.3 ± 6.9	60.1 ± 10.6	15.0	<0.001
% time spent standing	17.3 ± 5.6	21.1 ± 7.7	20.8 ± 5.5	20.3 ± 5.9	11.6	0.001
% time spent stepping	13.8 ± 5.4	21.9 ± 9.3	13.9 ± 4.0	19.6 ± 7.3	11.7	0.001
***ActiGraph data***						
% time spent in SB	50.8 ± 9.3	42.8 ± 13.3	51.9 ± 8.4	47.2 ± 10.7	20.4	<0.001
% time spent in LPA	45.5 ± 8.3	50.2 ± 10.9	44.2 ± 7.4	45.8 ± 7.3	34.6	<0.001
% time spent in MVPA	3.7 ± 2.5	7.0 ± 6.2	3.9 ± 2.3	7.0 ± 6.9	0.2	0.621

SD: Standard Deviation; SB: Sedentary Behaviour; LPA: Light Physical Activity; MVPA: Moderate to Vigorous Physical Activity; F F-ratio derived by ANCOVA; p probability value; * Comparison of follow-up values between groups, weighted by school, with baseline values included as the covariate.

**Table 4 ijerph-18-04759-t004:** Data on sitting bouts derived by the activPAL and the effect of the treatment (intervention).

	Control		Intervention		ANCOVA * Group Effect	
	Baseline	Follow-Up	Baseline	Follow-Up	F	*p*
**During Class Time**						
Total sitting bout (*N*)	39.5 ± 16.8	40.5 ± 14.0	43.9 ± 12.2	43.6 ± 12.8	1.4	0.245
Short sitting bout (*N*)	37.7 ± 17.7	38.2 ± 14.9	41.8 ± 12.5	42.5 ± 13.1	5.3	0.022
Medium-to-long sitting bout (*N*)	1.2 ± 1	1.1 ± 1.1	0.8 ± 0.5	0.4 ± 0.5	112.7	<0.0005
% spent in short sitting bouts	75.0 ± 22.2	79.0 ± 19.2	81.3 ± 11.3	89.2 ± 13.8	39.8	<0.001
% spent in medium-to-long sitting bouts	25.0 ± 22.2	21.0 ± 19.2	18.7 ± 11.3	10.8 ± 13.8	39.8	<0.001
Sit-to-stand transition	38.9 ± 16.8	39.5 ± 14.1	43.2 ± 12.1	43.0 ± 12.8	2.2	0.135
**After School Time**	**Baseline**	**Follow-Up**	**Baseline**	**Follow-Up**	**F**	***p***
Total sitting bout (*N*)	47.0 ± 14.1	46.6 ± 15.8	41.8 ± 11.3	44.3 ± 10.1	0.3	0.557
Short sitting bout (*N*)	45.8 ± 14.8	45.8 ± 16.2	40.7 ± 11.6	43.5 ± 10.5	0.3	0.556
Medium-to-long sitting bout (*N*)	1.6 ± 0.8	1.2 ± 0.9	1.2 ± 0.7	1.3 ± 0.9	10.1	0.002
% spent in short sitting bouts	72.2 ± 15.3	76.4 ± 16.2	73.2 ± 15.8	74.0 ± 16.5	3.6	0.057
% spent in medium-to-long sitting bouts	27.8 ± 15.3	23.6 ± 16.2	26.8 ± 15.8	26.0 ± 16.5	3.6	0.057
Sit-to-stand transition	47.2 ± 14.2	46.8 ± 15.8	41.9 ± 11.3	44.7 ± 10.2	0.1	0.758

F F-ratio derived by ANCOVA; p probability value; * Comparison of follow-up values between groups, weighted by school, with baseline values included as the covariate.

## Data Availability

Data can be made available upon request.

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
