# Peer review of "Stand Out in Class: Investigating the Potential Impact of a Sit–Stand Desk Intervention on Children’s Sitting and Physical Activity during Class Time and after School"

_ijerph, 2021, doi:10.3390/ijerph18094759_

Round 1

Reviewer 1 Report

The paper titled “Stand Out in Class: Investigating the potential impact of a sit-stand desk intervention on children’s sitting and physical activity during class time and after school” written by Chen and colleagues, it is a work that associates the enriched academic environment and the physical activity. This work extends previous findings of the research team but additionally incorporates populations with different ethnic heritage. The present work is well done.

I include these comments that could be useful for the authors.

  1. In table 2, what meaning SB Time? 
  2. The step count is higher in post-school time than school time in both groups. Certainly, this observation is logical, but how much time spent on the pupils in the school vs home? and how this could affect the study? 
  3. Curiously, the baseline MVPA time was higher in the control group (table 2), then this parameter was improved in the intervention group, but the values it is closed in both follow-up groups, which are the p values of the statistical analysis?

Reviewer 2 Report

The present manuscript covers an interesting topic that aimed to provide a comprehensive understanding of potential impact of a classroom sit-stand desk intervention on children’s sitting and physical activity behaviors within the classroom and after school settings.

Manuscript:

  1. Avoid unnecessary abbreviations in your manuscript.  

Abstract:

  1. Avoid using abbreviations in the abstract. 

Introduction:

  1. The introduction is well-writing.

Methods:

  1. The methodology is clear and well-structured 

Results and Tables:

Table 1:

  • Groups should be statistically compared on variables such as age, sex, ethnicity or BMI.
  • Please, provide the p-values.
  • These data should be similar between groups.
  • Does the mean BMI classified them as eutrophic?
  • Please, describe the prevalence of overweight and obesity in each group.

Table 2:

  • The abbreviations should be included in the footnotes.
  • Both groups should be statistically compared on the baseline on variables such as sitting time, standing time, stepping time and other variables.

Table 3:

  • All the abbreviations should be included in the footnotes. 

Discussion:

  1. The discussion proceeds nicely.
  2. Please, discuss these missing statistical analysis.

Reviewer 3 Report

The study is generally well written. Please clarify on :

  1. Why were two physical activity monitors used in the study? Why weren't all outcome metrics derived from a single (ActiGraph or activPAL) physical activity monitor?
  2. What is the purpose of using Kinesoft software when ActiLife software also allows using custom cutpoints for activity intensity estimation?

This study investigates the impact of sit/stand desks in children’s classroom as an environmental intervention to increase non-sedentary time in children. Some strengths and areas of improvement of the study include:

  • This is a well-designed cluster RCT. The selection of the geographical area of schools is justified well. Schools having students from different ethnicities and socio-economic status are a part of this RCT, which further strengthens its design.
  • The introduction sets-up the study quite well. Literature is cited to establish the need for the specific study design of the study.
  • Outcome metrics derived from the two wearable devices are good. The two wearable devices chosen are popular devices in the physical activity research domain.
  • The analysis (statistics) performed are appropriate for the study.
  • The study results are discussed well.
  • While the study addresses the scope defined, the study has potential for more in-depth analysis to be performed (for example: Investigate the differences in physical activity levels for students/schools that differ based on ethnicity/ socio-economic status within the intervention group). However, the scope defined for the current analysis is limited.
  • Some specifics of the methods such as the reasoning for using two different wearable devices and the need for Kinesoft software needs to be provided.
